# Study for Recycling Water Treatment Membranes and Compnents towards a Circular Economy—Case of Macaronesia Area

**DOI:** 10.3390/membranes12100970

**Published:** 2022-10-02

**Authors:** Tomás Tavares, Federico Leon, Jenifer Vaswani, Baltasar Peñate, Alejandro Ramos-Martín

**Affiliations:** 1Faculdade de Ciências e Tecnologia e da Escola de Ciências Agrárias e Ambientais, University of Cape Verde, Praia CP 279, Cape Verde; 2Departamento de Ingeniería de Procesos, Universidad de las Palmas de Gran Canaria, 35017 Las Palmas de Gran Canaria, Spain; 3Departamento de Aguas, Instituto Tecnológico de Canarias, 35019 Santa Lucía, Spain

**Keywords:** water treatment, reverse osmosis, membranes, desalination, recycling

## Abstract

Desalination is an opportunity to get fresh water for irrigation and for drinking. Reverse Osmosis (RO) for sea water desalination is a solution for the high demand for water in Atlantic islands. The most efficient process to get desalinated water is RO; however, it is necessary to study what to do with the RO membranes used at the end of their life. This paper confirms the possibility to recycle them. The main categories of recycling by thermal processing commonly used in the industry include incineration and pyrolysis to produce energy, gas and fuel. These processes can be applied to mixed plastic waste, such as the combination of materials used in the manufacture of RO membranes. Recycling RO elements from desalination plants is shown to be an opportunity and pioneering initiatives are already underway in Europe. Energy recovery, via incineration, is feasible nowadays and it is a possibility to recycle RO membranes. On the other hand, the recycling of RO elements, via the pyrolytic industry, for fuel production could be centralized in a new industry already planned in the Macaronesia area and all obsolete osmosis membranes could be sent there for recycling. Recycling RO membranes is a very important opportunity for the environment and economy of the zone. This is a new business in water treatments with membranes, very interesting for decreasing the residues and the carbon footprint. The importance of this work is applied to sea water membranes, brackish water ones, and also wastewater tertiaries RO elements at the end of their life.

## 1. Introduction

Water is a necessary resource for human activities, economic development and social welfare [1,2,3]. It is estimated that 50% of the world’s population will live in regions with a lack of water in 2025; hence the importance of proper water management and treatment [4]. According to the World Health Organization (WHO), irrigation is the activity that consumes the freshest water in the world, which represents about 70% of all global water withdrawals [5]. When water resources are scarce, its use in agriculture is limited, so it is necessary to look for alternatives such as Reverse Osmosis (RO) desalination to obtain such water [6].

By means of RO, a solvent can be separated with high efficiency from the solutes it carries dissolved, obtaining permeates with a salt concentration in the order of 1–5% of the final concentration. Semi-permeable membranes, which selectively allow the solvent to pass through while retaining salts, are a key element in the process. Initially, they were made of cellulose acetate, but recently polyamide membranes have proven to be more efficient by facilitating control of pore size and permeability. In general, RO membranes are very poorly permeable to ions and electrostatically charged particles, while they show very little resistance to the passage of dissolved gases (oxygen, carbon dioxide, etc.) and to electrostatically uncharged molecules of low molecular weight. A key operating parameter of RO which can pose a threat is membrane fouling. This is due to causes such as salt precipitates present in the feed and which in the concentrate can exceed the solubility product, sediments formed by colloids and other suspended particles, and finally, the growth of micro-organisms on the membrane surface [2,3,4].

The membrane cleaning technique will depend on the type of membrane and the main cause of fouling. Thus, in general, periods of membrane rinsing, in which cleaning solutions are circulated at high speed over the membrane surface, are alternated with periods in which the membranes are immersed in cleaning solutions. A good cleaning program extends the life of the membranes [5,6,7,8].

These are usually:To remove salt precipitates, an acid solution (hydrochloric, phosphoric or critical acid) and agents such as EDTA.To remove sediments and organic compounds, alkaline solutions combined with surfactants.To eliminate micro-organisms, chlorine solutions and chlorine derivatives to sterilize membranes.

This paper is related to recycling the RO membranes as a solution for these elements at the end of their life. Therefore, this study takes into account the circular economy [1,2,3].

The document studies the possibility to recycle the RO membranes. The main categories of recycling by thermal processing commonly used in the industry include incineration and pyrolysis to produce energy, gas and fuel. These processes can be applied to mixed plastic waste, such as the combination of materials used in the manufacture of RO membranes [6,7,8,9].

The recycling of RO elements from desalination plants is shown to be an opportunity and pioneering initiatives are already underway in Europe. Energy recovery, via incineration, is feasible but is not considered in line with the environmental, social, and political problems it may generate [10,11,12,13,14].

However, the recycling of RO membranes via the pyrolytic industry for fuel production can be centralized in a new industry already planned in the Macaronesia area and all obsolete osmosis membranes can be sent there. MITIMAC and Desal + projects promote the mitigation of climate change through innovation in the water cycle using low carbon technologies (www.mitimac.com, accessed on 9 August 2022) [10,15,16,17,18,19].

Recycling obsoleted RO elements is a very important opportunity for the environment and economy, decreasing the residues and the carbon footprint. The importance of this work could be extended to all water treatment membranes by RO (brackish water, sea water and wastewater elements) but also for NF or UF membranes at the end of their life [15].

## 2. Technical Actions to Be Carried Out

### 2.1. State-of-the-Art Recycling of Reverse Osmosis Membranes in Maccaronesia

Regarding membrane recycling in the Atlantic Islands, in the Canary Islands, there are 330 desalination plants in operation, concentrated on the islands of Gran Canaria, Tenerife, Fuerteventura and Lanzarote, where 291 desalination plants use RO technology (Table 1) [5,20,21,22,23,24].

A study of the number of elements in use and their replacement per year on these islands has been carried out. It is worth mentioning that the recycling of RO membranes does not exist in the Canary Islands. A German company that carries out this procedure recycles these membranes, but the transport costs must be considered, so it is more economical to take them directly to the landfill [6,25,26,27,28].

There are two types of procedures for recycling membranes: the first, mentioned above, is to reuse obsolete membranes in multi-stage plants that can be combined with membranes, putting the oldest membranes in the last stage or through the pyrolysis process [2,3,4,5].

#### 2.1.1. Types of RO Membranes

Semi-permeable membranes selectively allow the solvent to pass through while retaining salts and are a key element in the process [3,4,5]. There are two types of RO membranes:

Cellulose acetate membranes: These are the first membranes to be made from this material. It has high permeability and selectivity of substances, is obtained at a low cost and the absorption of protein molecules is low. These membranes are considered ideal for filtration processes in aqueous solutions as well as in culture media and sera because they improve solvent flux and reject salts, making solutions more sterile.

This type of membrane is used for large flow rates. The more asymmetric the membrane, the better the solvent flux. It must be considered that bacteria can damage the membrane, so a pre-treatment with chlorine is necessary to avoid damage.

Polyamide membranes: These are made by combining two fused polymers on a fabric backing suitable for medium flow rates and perform better than cellulose acetate membranes by allowing control of pore size and permeability.

The advantages are that they have better thermal, chemical, and mechanical stability and are not affected by bacteria.

On the downside, they have low permeability and absorption problems. It must be considered that the pores of the filter membranes are very small and allow the passage of completely pure solute [3,4,5].

#### 2.1.2. Membrane Materials

The membrane material must be stable over a wide pH and temperature range, moreover it should have good mechanical integrity, and the lifetime of the membrane depends on this.

The performance of the membrane (permeability, salt rejection) is intended to be stable over a period under conditions of use, which for current commercial membranes is 5 to 10 years [4,5,6].

There are two main groups of polymeric materials used to produce commercial RO membranes: cellulose acetate and polyamide, which are most supported by polysulphone. In Figure 1, the chemical structures of the cellulose acetate, aromatic polyamide and polysulfone are shown [4,5,6].

In the case of cellulose acetate membranes, they hydrolyze rapidly at extreme pH. Therefore, the operating pH range for this membrane is 6 to 8. However, it has sufficient tolerance to free chlorine to allow operation with chlorinated feed water, as well as in-line disinfection to control bacterial growth. On the other hand, polyamide composite membranes are stable over a wider pH range but are susceptible to oxidative degradation by free chlorine. Membranes made from polysulphone, on the other hand, have excellent chemical resistance, which allows long periods of operation and cleaning at high chlorine contents (200 mg/l), as well as a wide pH range (1–13) [4,5,6].

The membrane composition is shown in Figure 2, including the fiberglass housing which is involving the RO element; the membrane sheets made of aromatic polyamide, polysulfone and polyester; plastic permeate spacers separating both sides of the flat sheet membrane, altogether creating the membrane envelope; the plastic feed spacers between the membrane envelopes, the permeate tube made by PVC; plastics endcaps; and bonded parts [15].

In this paragraph, other information including chemical tolerance, membrane manufacturing method for testing, operating conditions, and performance of RO elements is shown.

The membrane type is cross-linked with fully aromatic polyamide composite. The test conditions of the manufacturing of these elements are the following:

Feed Water Pressure: 5.52 MPa;

Feed Water Temperature: 25 °C;

Feed Water Concentration: 32,000 mg/L NaCl;

Recovery Rate: 8%;

Feed Water pH: 7.

For more information, the minimum salt rejection for this type of RO membrane is approximately around 99.5%; and Boron rejection is about 95% at pH 8 (5 mg/L Boron added to Feed water) [1,15].

Regarding the operation limits of these RO elements, they are the following:

Maximum Operating Pressure: 8.3 MPa;

Maximum Feed Water Temperature: 45 °C;

Maximum Feed Water SDI15: 5;

Feed Water Chlorine Concentration: Not detectable;

Feed Water pH Range, Continuous Operation: 2–11;

Feed Water pH Range, Chemical Cleaning: 1–12;

Maximum Pressure Drop per Element: 0.10 MPa;

Maximum Pressure Drop per Vessel: 0.34 MPa.

Moreover, for more information, all the RO elements are wet tested, treated with a tested feed water solution, and then vacuum packed in oxygen barrier bags with deoxidant inside. To prevent biological growth during system shutdown, it is recommended to perform 30–60 min flushing of RO elements with feed or permeate water once every two days [1].

The presence of free chlorine and other oxidizing agents under certain conditions, such as heavy metals which act as an oxidation catalyst in the feed water will cause unexpected oxidation of the membrane. It is strongly recommended to remove these oxidizing agents contained in feed water before operating the RO system [1,15].

#### 2.1.3. Recycling of RO Membranes

Recycling consists of converting waste into new products or raw materials for subsequent use rather than disposal. In the Canary Islands, there is a high production of membrane waste that ends up in landfill sites and which represents a very high cost, so these elements can give a second life. For example, brackish water RO membranes can be given a new life in other plants which do not need a high permeate quality. However, this water use would only be for irrigation or for return to the sea because the water from the WWTP to the tertiary is very dirty compared to the water from a brackish well installation [2,3,4,5,6].

For the recycling of the membranes, the best membranes will be chosen with a chemical cleaning procedure in order to be able to reuse them in the treatment plant, thus extending their useful life. However, in multi-stage plants, membranes can be combined with membranes, placing the oldest membranes in the last stage, thus precipitating the salts earlier and using them to produce products with these salts [2,3,4,5,6].

The advantages that we can find in terms of recycling is that we will be favoring the elimination of membrane waste, so we will avoid a high cost, as each membrane is around 500 euros, so if the membrane has a useful life of 5–7 years, this means an annual amortization cost of 100 euros for each membrane out of a total of 49,500 membranes operating in the Canary Islands [2,3,4,5,6].

As for the disadvantages, we can find membranes that have high fouling and cannot be reused, for this, a pre-treatment is required prior to the membranes to avoid clogging with solids or precipitates [2,3,4].

Another disadvantage is that the desired performance would not be obtained and could be affected due to the use of used membranes, as they would not have the same efficiency as a new one [2,3,4].

The RO composite membrane sheet is usually constructed with a thin dense layer of polyamide, supported by an inner layer of microporous polysulphone (PSf) and a non-woven polyester [2,3,4,5,6].

The large quantity of obsoleted RO membranes from desalination of the Macaronesia area is really a global problem for the Atlantic islands as we do not know what to do with them and it is a problem for the environment in small islands and extensions. However, the recycling of RO membranes is a social opportunity for the environment and financial issues too, in the Macaronesia area, Spain and in Europe. In fact, in Germany, there is a company that is recycling membranes with a price of around 250 EUR per 10 elements recycled, which is very interesting for the circular economy and society. Moreover, the recycling process of RO membranes and the pyrolytic industry are technically viable and potential opportunities in the actual industry of the recycling market [2,3,4,5,6].

Recycling RO membranes offers technical and economic advantages. The main results of this project will help minimize environmental impacts by increasing the life cycle of the membrane elements through secondary use or reuse of materials and, therefore, reduce the carbon footprint and further improve the sustainability of the technology [2,3,4].

#### 2.1.4. Pyrolysis Process

Pyrolysis is the chemical decomposition of organic matter and all types of materials (except metals and glass) caused by heating at high temperatures in the absence of oxygen. In this pyrolysis process, residue of liquid, gas and a solid residue are obtained. Gaseous and liquid waste could be utilized by combustion through a steam cycle to produce electrical energy. Solid waste can be used as fuel in industrial installations [6,7,8].

In order to carry out the energy assessment, it is necessary to eliminate glass fiber encapsulation that provides the membrane [image 1.2], the only disadvantage being the other elements due to the difficulty in separating them because they are mixed together, being plastic membranes. Although, with pyrolysis being the thermal treatment, mixing several types of plastics, we can finally obtain fuels with high calorific value as by-products of these elements [6,7,8].

Now, the disadvantage is that there is no incinerator in the Canary Islands, but Tenerife has launched a project called “Plastics Energy”, where all the RO elements for this process could be sent, in case it is not carried out, this project could be centralized in any other island for the same purpose. These are the parts of the RO membranes to be used in the pyrolysis process as shown in Figure 3 and Figure 4 below.

The different components of the RO membranes and their characteristics, shown in Figure 5 and Figure 6, are the following. In the flat sheet membranes, we found the aromatic polyamide material, a polysulphone layer and polyester. Moreover, the polypropylene feed spacer is in between the flat sheet membranes, mixed with glue. Finally, it is made of ABS or noryl materials, such as the end caps and permeate tubes.

#### 2.1.5. Other Options for Disposal

In terms of options for disposal, other strategies may be considered in the future. For example, sheets and spacers have previously been recycled as geotextiles in home gardens under a layer of gravel to maintain the position of decorative rocks and eliminate weed growth [5]. In addition, potential agricultural applications for spacers have also been proposed, including bird netting, windbreaks, or turf protection netting [2,3,4,5,6].

## 3. Results and Discussions

In the investigation, we found around 50,000 RO membranes in operation in Macaronesia territories and a partial replacement per year of about 10,000 membranes, around 20% of all [13,17,27].

In our estimation, in the Atlantic islands, the membranes are used for at least 5 years, being that the standard warranty per element by the manufacturer is 3 years. Usually, obsoleted membranes go to landfills for burying. For more information, you can find the below documentation used for this study about the SWRO plants database of the Canary Islands. This is shown in Table 2 and Figure 7 and Figure 8 [28,29,30,31,32].

Regarding the total permeate flow in the Macaronesia area, estimated at 687,000 m^3^/d. In Table 3, the number of elements used per year, partial replacement per year, their weight and volume are shown [13,17,27,32].

In the process of incineration, the encapsulated glass fiber should be eliminated from the elements. This is not complicated to do. Other parts of the membranes are difficult to eliminate because they are mixed with glue and plastic elements; however, we can incinerate them to produce energy [28,29,30,31,32,33,34].

In the Macaronesia area, the incinerator could be centralized on one island, and all the obsoleted RO elements could be sent there. On Tenerife island, there is a pyrolytic where the RO membranes can be treated in this way [32,35].

This pyrolytic industry is working in between 300 °C and 800 °C. It is obtained in the waste from this process recycled components such as liquid and solid residues, and the following gases: CH_4_, H_2_, CO_2_ and CO [27,32,36].

Next, the financial-economic plan is described to evaluate the economic viability of the results of this study.

RO membrane recycling does not exist in the Canary Islands and Macaronesia area; however, the German company MEMRE is doing it, and it has a cost of 25 Euros per reverse osmosis membrane plus transport. Therefore, it is possible to also do it in the Atlantic islands [32,37].

In the Canary Islands, at this time, between 20 and 30 Euros are paid plus transport to take the reverse osmosis elements to a landfill to bury them without any treatment, in a similar way in all the islands. In other autonomous communities, it is also taxed, for example, in Alicante the reverse osmosis membranes are evacuated from the plant as normal urban solid waste (RSU) removal and have a cost of 45 Euros/Tm in management plus taxes, in addition, they have to add transportation. Energy recovery or fuel production through pyrolysis are two recycling options that would avoid these segregation costs [13,17,27,38,39,40].

An estimated economic analysis has been studied for the operating account of the new potential recycling business of reverse osmosis membranes and their components, also for a period of 5 years, to recycle them in other processes such as incineration or pyrolysis. The initial business cost has been estimated at around EUR 80,000 to have the necessary facilities to be able to manage the recycling of the membranes, that is, office, staff, etc., since it is not intended to build an incinerator or a pyrolytic industry in this case but to take advantage of the synergies that may arise with the future projected pyrolytic industry in Tenerife “Plastics Energy” [32,40].

In the same way, personnel expenses (salaries) have been increased annually with an estimate of 3% of the CPI, amortizations over a time horizon of a minimum of 20 years of duration of the activity, the initial number of membranes as studied in part of 10,000 obsolete elements generated annually and it is estimated that each one of them can be received at least 10 EUR, which is what has been considered in the calculations [27,32,39].

Finally, the following estimate is obtained, reaching an internal rate of return of 73%, which, taking into account that the current interests of the European Bank are 0% and that we would also save the cost of taking the obsolete membranes to the landfill or to the Competent authorized manager according to the autonomous community where we are located is also a viable business opportunity both technically and economically and very interesting in the market. You can find this information in the following Table 4 [39,40].

This economic analysis justifies the recycling of the obsoleted RO elements and obtained an interest rate of 73%, which is very high and interesting for any business. Therefore, companies such as Plastics Energy are introduced in this market. Due to the good economic results of the study this organization is constructing a pyrolytic industry in Tenerife Island, the biggest one in the Macaronesia area to centralize all this business in the zone [15,32].

The recovery of plastics, both material and energy, is going to be vital in the near future and, given that the Administration is focused on waste of domestic and commercial origin, it will be up to the private sector to take the initiative, and try together with the Administration that there is no unfair competition by dumping waste of this type, since it is currently much cheaper to do so and the implementation of other more environmentally sustainable technologies will not be favored if it is not promoted, which is why a very interesting line emerges in this sense and it is possible to create a project for the development of this route [15,32].

Moreover, in Appendix A, more information on the fixed, variable and personnel costs of a typical desalination plant of 50,000 m^3^/d according to the FCCA and which has been taken as a reference for this study is shown.

## 4. Conclusions

The recycling of RO membranes is a very important opportunity for the environment and economy, specifically in the Macaronesia area, as shown in Table 2 of the economic analysis performed to justify these conclusions.

Recycling obsoleted RO elements in desalination is a new business in water treatments with membranes. It is very interesting for the environment, decreasing the residues and the carbon footprint. In fact, the importance of this work is applied to sea water membranes, brackish water ones, and also wastewater tertiaries RO elements at the end of their life.

The main categories of recycling by thermal processing commonly used in the industry include incineration and pyrolysis to produce energy, gas and fuel.

One of the recycling processes is pyrolysis. In this process, energy and recycled components are obtained from the old RO elements of the desalination plants. This could be applicable in the Atlantic Islands which have a high number of desalination plants per surface.

On the other hand, incineration is also an interesting process to consider for recycling the obsoleted membranes to produce energy.

## Figures and Tables

**Figure 1 membranes-12-00970-f001:**
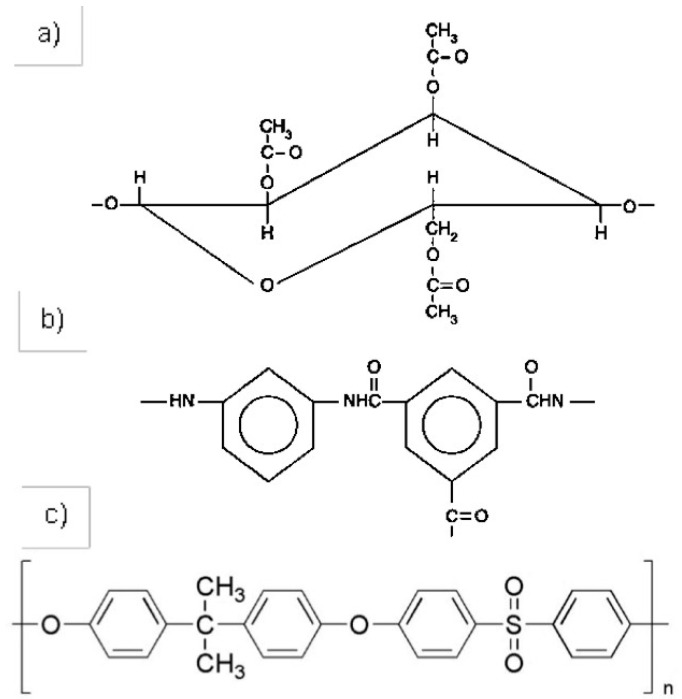
Chemical structure of: (**a**) Cellulose acetate; (**b**) Polyamide; (**c**) Polysulfone. Source: self-made.

**Figure 2 membranes-12-00970-f002:**
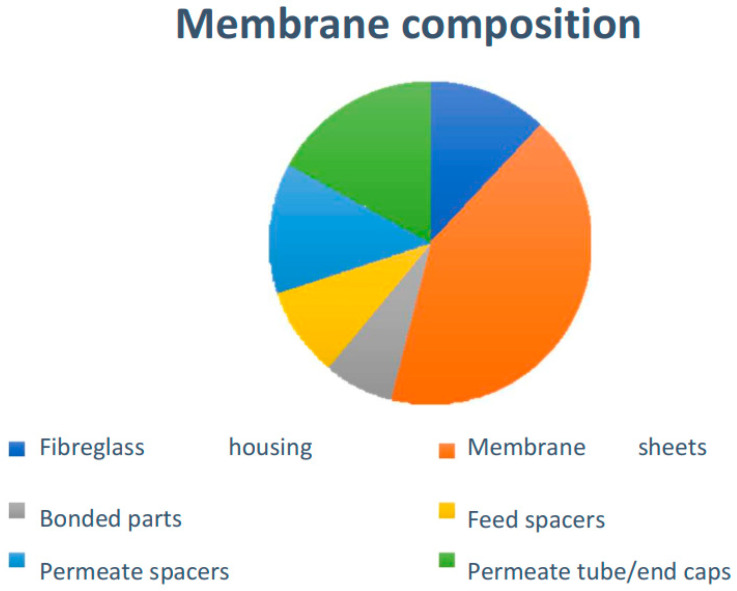
Membrane composition. Source: self-made.

**Figure 3 membranes-12-00970-f003:**
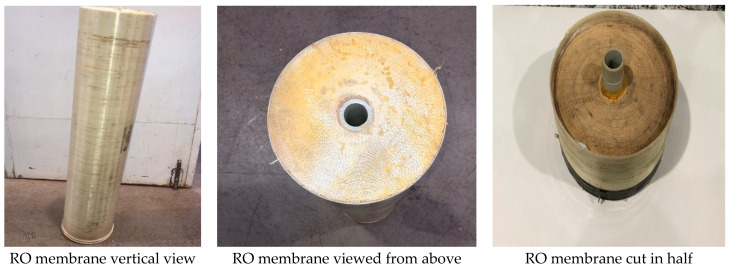
RO Membranes. Source: self-made.

**Figure 4 membranes-12-00970-f004:**
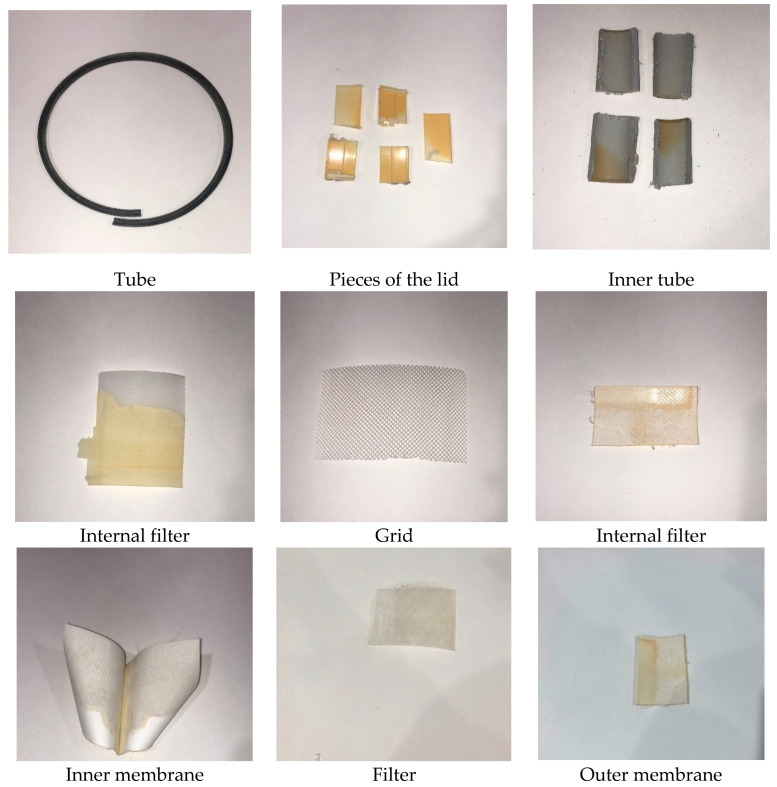
Name of the components of the I.O.M. Source: self-made.

**Figure 5 membranes-12-00970-f005:**
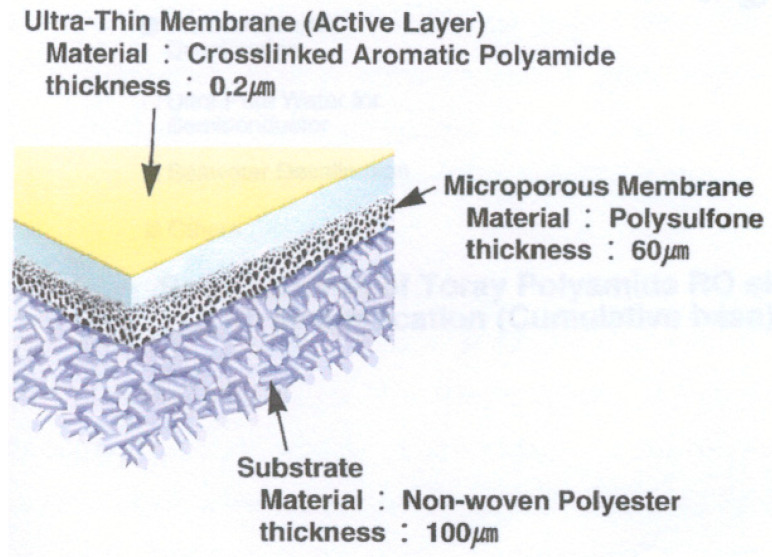
Flat sheet membrane section [1].

**Figure 6 membranes-12-00970-f006:**
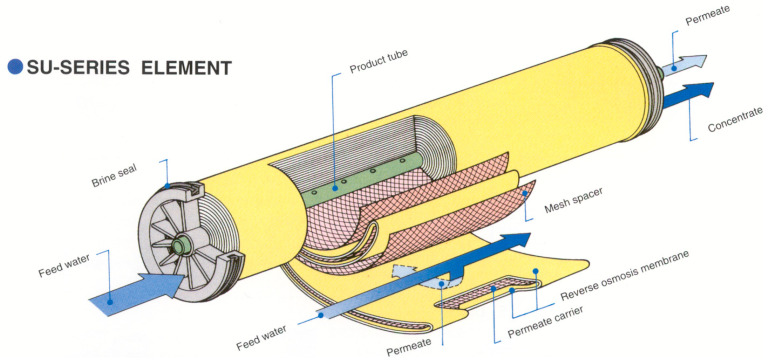
RO element section [1].

**Figure 7 membranes-12-00970-f007:**
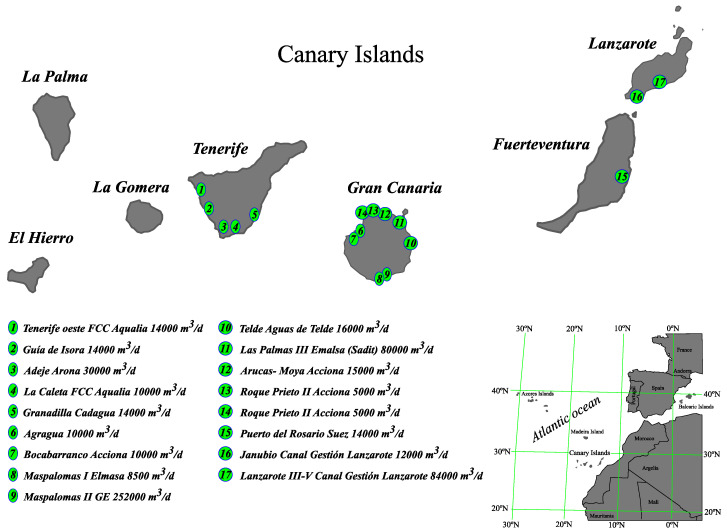
Most significant sea water desalination plants (2019) [15,32]. Source: self-made.

**Figure 8 membranes-12-00970-f008:**
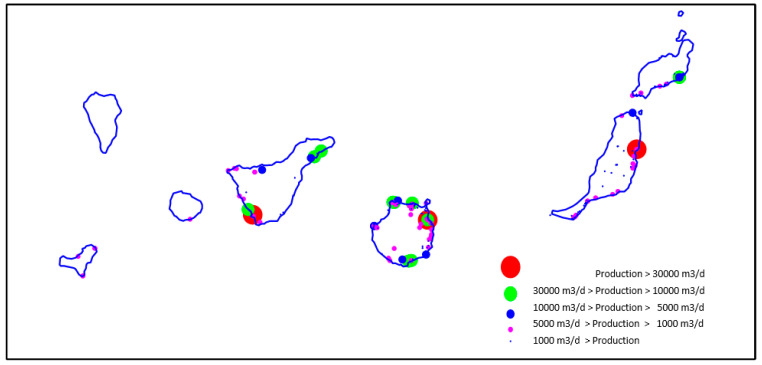
Most significant sea water desalination plants in the Canary Islands (Spain) (2019) [15,32]. Source: self-made.

**Table 1 membranes-12-00970-t001:** Number of membranes and replacement per year. Source: self-made.

Island	Nº. of Elements/Year Used	Element Replacement/Year
Gran Canaria	23500	4700
Tenerife	9350	1870
Fuerteventura	8650	1730
Lanzarote	8000	1600

**Table 2 membranes-12-00970-t002:** SWRO Plants in Canary Islands [15,32]. Source: self-made.

Name of the Plant	Production (m^3^/d)	Consume (kWh/m^3^)	Island	Habitants per Plant
Cercado de Don Andrés	200	3.5	Lanzarote	Irrigation
Lanzarote III 1	10,000	3.5	Lanzarote	10541
Lanzarote III 2	5000	3.5	Lanzarote	5271
Lanzarote III 3	5000	3.5	Lanzarote	5271
Lanzarote IV	20,000	3.5	Lanzarote	21083
Lanzarote V	18,000	2.4	Lanzarote	18975
Aeropuerto	700	3.04	Lanzarote	18327
Agua Park	30	3.04	Lanzarote	500
Apartamentos Ficus	60	3.5	Lanzarote	120
Apartamentos Puerto Tahiche	150	3.5	Lanzarote	300
Apartamentos Trebol	80	3.5	Lanzarote	160
Ercros	2500	3.5	Lanzarote	11057
Ercros	2200	3.5	Lanzarote	9731
Famara	350	3.5	Lanzarote	700
Hotel Golf y Mar	90	3.5	Lanzarote	180
Hotel Gran Meliá Salinas	400	2.61	Lanzarote	800
Hotel Playa Verde	250	3.5	Lanzarote	500
Hotel Teguise Playa	250	3.5	Lanzarote	500
La Galea	150	3.04	Lanzarote	300
Lanzarote Beach Club II	70	3.04	Lanzarote	140
Las Arenas. Costa Teguise	80	3.04	Lanzarote	160
Playa Roca	250	3.04	Lanzarote	500
Apartamentos Don Paco Castilla	320	2.61	Lanzarote	640
Apartamentos Sol Lanzarote	350	2.61	Lanzarote	700
Cdad Apartamentos CAMP		2.61	Lanzarote	Tourism
Holiday Land S.A.	3000	3.5	Lanzarote	6000
Hotel Fariones Playa	500	3.5	Lanzarote	1000
Hotel Playa Azul	300	3.5	Lanzarote	600
Hoteles Canarios S.A.		3.5	Lanzarote	Tourism
Iberhotel		3.5	Lanzarote	Tourism
Zorilla	40	3.04	Lanzarote	80
Hotel Jameos Playa	336	2.61	Lanzarote	672
La Santa Sport I	250	3.5	Lanzarote	500
La Santa Sport II	250	3.5	Lanzarote	500
Ria La Santa	400	3.5	Lanzarote	800
Apartamentos Son Boy Family Suites	500	3.04	Lanzarote	1000
Bungalows Atlantic Gardens		3.5	Lanzarote	Tourism
Costa los Limones S.A.	350	3.5	Lanzarote	700
Hotel Corbeta		3.5	Lanzarote	Tourism
Hotel Costa Calero	324	3.04	Lanzarote	642
Marina Rubicón	300	3.04	Lanzarote	600
Hotel Paradise Island	300	3.04	Lanzarote	600
Hotel Princesa Yaiza	500	3.04	Lanzarote	1000
Hotel Rubicón Palace	450	3.04	Lanzarote	900
Inalsa Sur 1	600	3.5	Lanzarote	1859
Inalsa Sur 2	1200	3.5	Lanzarote	3718
Inalsa Sur 3	3000	3.5	Lanzarote	9294
Janubio		3.04	Lanzarote	Tourism
Lanzasur Club	200	3.04	Lanzarote	400
Playa Blanca S.A.		3.5	Lanzarote	Tourism
Club Lanzarote	4500	3.5	Lanzarote	9000
Apartamentos Moromar	250	3.5	Lanzarote	500
Gea Fonds Numero Uno Lanzarote S.A.		3.5	Lanzarote	Tourism
Grupo Rosa	1000	3.5	Lanzarote	2000
Hipotels	300	3.5	Lanzarote	600
Hotel Corona	300	3.5	Lanzarote	600
Hotel Costa Calero S.L.	300	3.04	Lanzarote	600
Hotel Sunbou	500	3.04	Lanzarote	1000
Isla Lobos	100	3.04	Lanzarote	200
Leas Hotel S.A.		3.5	Lanzarote	Tourism
Niels Prahm		3.5	Lanzarote	Tourism
Occidental Hotel Oasis	250	3.04	Lanzarote	500
Playa Flamingo	200	3.04	Lanzarote	400
Tjaereborg Timesharing. S.A.	500	3.04	Lanzarote	1000
Empresa Mixta de Aguas de Antigua. S.L.	4800	3.04	Fuerteventura	11948
Grupo Turístico Barceló. S.L.	240	3.5	Fuerteventura	480
Aguas Cristóbal Franquis. S.L.	1200	3.5	Fuerteventura	2400
Anjoca Canarias. S.A.	3000	3.5	Fuerteventura	6000
Ramiterra. S.L.	3000	3.04	Fuerteventura	6000
Inver Canary Dos. S.L.	300	3.04	Fuerteventura	600
Suministros de Agua de La Oliva. S.A.	9000	3.04	Fuerteventura	17920
Consorcio Abastecimiento de Aguas a Fuerteventura	4000	3.04	Fuerteventura	7964
Parque de Ocio y Cultura (BAKU) 1	300	3.04	Fuerteventura	600
Parque de Ocio y Cultura (BAKU) 2	90	3.04	Fuerteventura	180
RIU Palace Tres Islas	100	3.5	Fuerteventura	200
RIU Oliva Beach	400	3.5	Fuerteventura	800
Nombredo. S.L.	500	3.5	Fuerteventura	1000
Consorcio Abastecimiento de Aguas a Fuerteventura	4400	3.5	Fuerteventura	20539
Puertito de la Cruz	60	3.5	Fuerteventura	120
Vinamar. S.A.	3600	3.5	Fuerteventura	7200
Fuercan. S.L. Cañada del Rio I	2000	3.5	Fuerteventura	4000
Fuercan. S.L. Cañada del Rio II	1000	3.04	Fuerteventura	2000
Fuercan. S.L. Cañada del Rio III	2000	3.04	Fuerteventura	4000
Club Aldiana	200	3.5	Fuerteventura	400
Erwin Sick	30	3.5	Fuerteventura	60
Esquinzo Urbanización II	1200	3.5	Fuerteventura	2400
Esquinzo Urbanización III	1200	3.5	Fuerteventura	2400
Hotel Sol Élite Los Gorriones 1	400	3.5	Fuerteventura	800
Hotel Sol Élite Los Gorriones 2	400	3.5	Fuerteventura	800
Stella Canaris I	300	3.5	Fuerteventura	600
Stella Canaris II	300	3.5	Fuerteventura	600
Stella Canaris III	250	3.5	Fuerteventura	500
Hotel H 10 Playa Esmeralda.	250	3.5	Fuerteventura	500
Hotel “Club Paraíso Playa”	300	3.5	Fuerteventura	600
Urbanización Costa Calma.	110	3.5	Fuerteventura	220
Urbanización Tierra Dorada.	120	3.5	Fuerteventura	240
Zoo-Parque La Lajita.	1300	3.5	Fuerteventura	500
Apartamentos Esmeralda Maris	120	3.5	Fuerteventura	240
Hotel H10 Tindaya	280	3.5	Fuerteventura	560
Aparthotels Morasol	80	3.5	Fuerteventura	160
Consorcio Abastecimiento de Aguas a Fuerteventura	36,500	3.5	Fuerteventura	39382
Aeropuerto	500	3.5	Fuerteventura	15439
GranTarajal	4000	3.5	Fuerteventura	14791
Sotavento. S.A.	2925	3.5	Fuerteventura	5850
Arucas-Moya I	10,000	3.5	Gran Canaria	45419
Granja experimental	500	3.5	Gran Canaria	Irrigation
Granja experimental	500	3.5	Gran Canaria	Irrigation
Comunidad Fuentes de Quintanilla	800	3.04	Gran Canaria	Irrigation
Granja experimental	500	3.5	Gran Canaria	Irrigation
Gáldar-Agaete I	3000	3.5	Gran Canaria	16199
Gáldar II	7000	3.04	Gran Canaria	37799
Agragua	15,000	3.5	Gran Canaria	Irrigation
Guía I	5000	3.5	Gran Canaria	6962
Guía II	5000	2.61	Gran Canaria	6962
Félix Santiago Melián	5000	2.61	Gran Canaria	Irrigation
Las Palmas III	65,000	3.5	Gran Canaria	307545
Las Palmas IV	15,000	2.61	Gran Canaria	70972
BAXTER S.A.	100	3.5	Gran Canaria	200
El Corte Inglés. S.A.	300	3.5	Gran Canaria	3000
Anfi del Mar I	250	3.5	Gran Canaria	500
Anfi del Mar II	250	3.5	Gran Canaria	500
AQUALING	2000	3.04	Gran Canaria	4000
Puerto Rico	4000	3.04	Gran Canaria	8000
Puerto Rico I	4000	3.04	Gran Canaria	8000
Hotel Taurito	400	3.04	Gran Canaria	800
Hotel Costa Meloneras	300	3.04	Gran Canaria	600
Hotel Villa del Conde	500	3.04	Gran Canaria	1000
Bahia Feliz	600	3.5	Gran Canaria	1200
Bonny	8000	3.5	Gran Canaria	Irrigation
Maspalomas I Mar	14,500	3.5	Gran Canaria	19572
Maspalomas II	25,200	3.04	Gran Canaria	34016
UNELCO II	600	3.5	Gran Canaria	Industrial
Ayto. San Nicolas	5000	3.04	Gran Canaria	7608
Asociación de agricultores de la Aldea	5400	3.04	Gran Canaria	Irrigation
Sureste III	8000	3.5	Gran Canaria	133846
Aeropuerto I	1000	3.5	Gran Canaria	24791
Salinetas	16,000	3.5	Gran Canaria	102424
Aeropuerto II	500	3.5	Gran Canaria	12396
Hoya León	1500	3.5	Gran Canaria	Irrigation
Bco. García Ruiz	1000	3.5	Gran Canaria	Irrigation
Mando Aéreo de Canarias	1000	3.5	Gran Canaria	3000
UNELCO I	1000	3.5	Gran Canaria	Industrial
Anfi del Mar	1500	3.04	Gran Canaria	3000
Norcrost. S.A.	170	3.04	Gran Canaria	340
Adeje Arona	30,000	3.04	Tenerife	126728
Gran Hotel Anthelia Park		3.04	Tenerife	Tourism
La Caleta (Ayto. Adeje)	10,000	3.04	Tenerife	20000
UTE Tenerife Oeste	14,000	2.16	Tenerife	40000
Hotel Sheraton La Caleta		3.04	Tenerife	Tourism
Hotel Gran Tacande		3.04	Tenerife	Tourism
Hotel Rocas de Nivaria. Playa Paraíso		3.04	Tenerife	Tourism
Hotel Bahía del Duque. Costa Adeje		3.04	Tenerife	Tourism
Siam Park		3.04	Tenerife	Tourism
Tenerife-Sol S. A.		3.04	Tenerife	Tourism
Hotel Conquistador. P. de Las Américas		3.04	Tenerife	Tourism
Arona Gran Hotel. Los Cristianos		3.04	Tenerife	Tourism
Bonny S.A. Finca El Fraile.		3.04	Tenerife	Tourism
El Toscal. La Estrella (C. Regantes Las Galletas)		3.04	Tenerife	Tourism
Complejo Mare Nostrum. P. Las Américas		3.04	Tenerife	Tourism
Hotel Villa Cortés		3.04	Tenerife	Tourism
Buenavista Golf S.A.		3.04	Tenerife	Tourism
Rural Teno		3.04	Tenerife	Agrícola
Ropa Rent. S.A. (P.I. Güímar)		3.04	Tenerife	Industrial
Unelco	600	3.5	Tenerife	Industrial
I.T.E.R. Cabildo de Tenerife	14	3.5	Tenerife	Industrial
C.T. en P.I. de Granadilla		3.5	Tenerife	Industrial
Bonny S.A. Finca El Confital.		3.5	Tenerife	Irrigation
Polígono Industrial de Granadilla (portátil)		3.5	Tenerife	Industrial
UTE Desalinizadora de Granadilla	14,000	3.04	Tenerife	50146
Guia de ISORA Hoya de la leña		3.5	Tenerife	Tourism
Club Campo Guía de Isora. Abama		3.5	Tenerife	Tourism
Hotel Meliá Palacio de Isora. Alcalá.		3.5	Tenerife	Tourism
Loro Parque		3.5	Tenerife	Tourism
Santa Cruz I	20,000	3.04	Tenerife	204856
Recinto Portuario Santa Cruz (portátil)		3.04	Tenerife	Industrial
CEPSA	1000	3.04	Tenerife	Industrial
Hotel Playa la Arena		3.04	Tenerife	Tourism
Hotel Jardín Tecina	2000	3.04	La Gomera	4000
La Restinga	500	3.5	El Hierro	297
La Restinga	1200	3.04	El Hierro	712
El Cangrejo	1200	3.04	El Hierro	2478
El Cangrejo	1200	3.04	El Hierro	2478
El Golfo	1350	3.04	El Hierro	4093

**Table 3 membranes-12-00970-t003:** Number of elements used, replacement, weight and volume [15,32]. Source: self-made.

Macaronesia Islands	Number of Elements Used per Year	Partial Replacement per Year	Weight (kg)	Volume (m^3^)
Lanzarote (Canary Islands)	8000	1600	32,000	67
Fuerteventura (Canary Islands)	8650	1730	34,600	73
Gran Canaria (Canary Islands)	23,500	4700	94,000	197
Tenerife (Canary Islands)	9350	1870	37,400	79
El Hierro (Canary Islands)	350	70	1400	3
La Gomera (Canary Islands)	150	30	600	1
Porto Santo (Madeira)	450	45	900	2
Praia-Palmarejo (Cape Verde)	700	100	2000	4
St Vincent (Cape Verde)	350	50	1000	2
Salt (Cape Verde)	350	50	1000	2

**Table 4 membranes-12-00970-t004:** Economic analysis of RO membrane recycling study in EUR [15,32]. Source: self-made.

TRADING ACCOUNT	YEAR 0	YEAR 1	YEAR 2	YEAR 3	YEAR 4	YEAR 5
Bills		500	300	300	350	350
Variation of existences		0	0	0	0	0
Shopping		5000	3000	3000	3500	3500
External services		1000	500	500	600	600
Personal expenses		30,000	30,900	31,827	32,782	33,765
Amortization		4000	4000	4000	4000	4000
Financial expenses		100	100	100	120	120
Other expenses		350	250	200	200	200
Total		40,950	39,050	39,927	41,552	42,535
Income		100,000	100,000	100,500	121,200	121,800
Variation of existences		0	0	50	100	150
Sales (obsolete membranes)		10,000	10,000	10,050	10,100	10,150
Other operating income		0	0	0	0	0
Other income		0	0	0	0	0
Financial income		0	0	0	0	0
Total	−80,000	59,050	60,950	60,573	79,648	79,265
Internal Rate of Returns (IRR)						73%

## Data Availability

Not applicable.

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
