# Peer review of "Study for Recycling Water Treatment Membranes and Compnents towards a Circular Economy—Case of Macaronesia Area"

_membranes, 2022, doi:10.3390/membranes12100970_

Round 1

Reviewer 1 Report (New Reviewer)

Results of the study are very weak and needs to be expanded in this study

Discussion section of this study is almost absent in this study

English language needs to be modifies and professional edit

There are only 2 tables in results, no pictorial presentations?

There is no graphical presentation of the developed RO membranes and their characteristics?

Author Response

Thanks so much for your review of this manuscript and we are very satisfied with your comments about the topic of this one.

Results of the study are very weak and needs to be expanded in this study

You are right. We have done a big effort in this issue increasing to 8 pages the results of this study, being them stronger, understandable, and expanding it. In this way, we improved much a lot the results of this study following the kind advice of the reviewer.

Discussion section of this study is almost absent in this study

Ok, we introduced the discussion section together the results section increasing this to 8 pages, making a big effort to improve this part of the paper. Finally, we have gotten, a more extended section for results and discussion as you can see in the document modified, introducing new references updated and discussed.

English language needs to be modifies and professional edit

Ok, we will send it to an English native speaker, specialist in scientific articles, to improve the language, since we get the approval for publication from the editor.

There are only 2 tables in results, no pictorial presentations?

You are right. Due to this and following the kind support of the reviewer, we have increased the number to tables in results and we have introduced pictorial presentations. You can find now in the article 4 tables and 8 figures explaining much better the results of this study.

There is no graphical presentation of the developed RO membranes and their characteristics?

Ok, we have introduced the new figures 3, 4, 5 and 6 including the components and characteristics of the developed RO membranes. Moreover, we have also explained this and introduced al these figures in the text.

Thanks so much again for your opinion in this way and strong support to improve a lot our manuscript and to publish it.

Reviewer 2 Report (New Reviewer)

The topic of “Study for recycling water treatment membranes and components towards a circular economy. Case of Macaronesia area” is an interesting topic, not only for Micronesia but globally, and even for the marine transport industry.

I think the chapter lacks a systematic organization, starting from the abstract. In fact, statements in the abstract contradict some of the conclusions within the article.

The authors should provide an overview of the global problem and its relevance to Micronesia. Since the article pertains to the recycling of membranes in that region, what specific technical and economic advantages are available? The authors should specify to make a case for the circular economy.

Also, the authors should do a bit more search in terms of the number of existing plants and their capacities. The data is significantly higher than mentioned in the article.

The figures should be placed closest to the place where they are mentioned in the article. Also, I also feel that figures are not indicative of the context.

The data on circular economy is probably extracted from literature and hence authors should specify the source.

The authors should add some latest references as well.

Overall, I think, the authors should rewrite the entire manuscript in a systematic and organized manner.

That said, I think, the research topic is very relevant and it is important to carry out systematic research.

Author Response

Author's Notes to Reviewer.

The topic of “Study for recycling water treatment membranes and components towards a circular economy. Case of Macaronesia area” is an interesting topic, not only for Micronesia but globally, and even for the marine transport industry.

Thanks so much for your review of this manuscript and we are very satisfied with your comments about the topic of this one.

I think the chapter lacks a systematic organization, starting from the abstract. In fact, statements in the abstract contradict some of the conclusions within the article.

Ok, we changed the organization of the paper, including the abstract, and also the conclusions to have all in accordance. Following the comments of the reviewer we could improve much more the manuscript. We have modified the abstract, conclusions and the organization of the document to follow up the instructions of the reviewer and thanks to this we could improve this a lot.

The authors should provide an overview of the global problem and its relevance to Micronesia. Since the article pertains to the recycling of membranes in that region, what specific technical and economic advantages are available? The authors should specify to make a case for the circular economy.

The big quantity of obsoleted RO membranes from desalination of Macaronesia area is really a global problem for the Atlantic islands due to we do not know what to do with them and it is a problem for the environment in small islands and extensions. However, the recycling of RO membranes is a social opportunity for the environment and financial issues too, in Macaronesia area, Spain and in Europe. In fact, in Germany there is a company which is recycling membranes with a price around 250 EUR per 10 elements recycled, which is very interesting for the circular economy and society. Moreover, the recycling process of RO membranes and the pyrolytic industry are technically a viable and a potential opportunity in the actual industry of the recycling market. Recycle RO membranes gives technical and economic advantages. For instance, this will help to minimise environmental impacts by increasing the life cycle of the membrane elements through secondary use or reuse of materials and, therefore, to reduce the carbon footprint and further improve the sustainability of the technology. An example is the case of Canary Islands where you can see in table 2 the high number of obsoleted RO elements produced per year which it is possible to recycle, to produced energy and recycled materials from them. For instance, using pyrolysis process for recycling, residue of liquid, gas and a solid residue are obtained. Gaseous and liquid waste could be utilised by combustion through a steam cycle to produce electrical energy. Solid waste can be used as fuel in industrial installations.

About more economic advantages and cases like Canary Islands and others we have introduced in the paper the following. RO membranes recycling does not exist in the Canary Islands and Macaronesia area, however the German company MEMRE is doing it, and it has a cost of 25 Euros per reverse osmosis membrane plus transport. Therefore, this is possible to do it also in Atlantic is-lands.

In the Canary Islands at this time between 20 and 30 Euros are paid plus transport to take the reverse osmosis elements to a landfill to bury them without any treatment, in a similar way in all the islands. In other autonomous communities it is also taxed, for ex-ample, in Alicante the reverse osmosis membranes are evacuated from the plant as normal urban solid waste (RSU) removal and have a cost of 45 Euros/Tm in management plus taxes, in addition there are to add transportation. Energy recovery or fuel production through pyrolysis are two recycling options that would avoid these segregation costs.

An estimate has been studied for the operating account of the new potential recy-cling business of reverse osmosis membranes and their components, also for a period of 5 years, to recycle them in other processes such as incineration or pyrolysis. The initial business cost has been estimated at around EUR 80,000 to have the necessary facilities to be able to manage the recycling of the membranes, that is, office, staff, etc., since it is not in-tended to build an incinerator or a pyrolytic industry in this case but to take advantage of the synergies that may arise with the future projected pyrolytic industry in Tenerife "Plas-tics Energy".

In the same way, personnel expenses (salaries) have been increased annually with an estimate of 3% of the CPI, amortizations over a time horizon of a minimum of 20 years of duration of the activity, the initial number of membranes as studied in part of 10,000 ob-solete elements generated annually and it is estimated that each one of them can be re-ceived at least 10 EUR, which is what has been considered in the calculations.

Finally, the following estimate is obtained, reaching an internal rate of return (IRR) of 73%, which, taking into account that the current interests of the European Bank are 0% and that we would also save the cost of taking the obsolete membranes to the landfill or to the Competent authorized manager according to the autonomous community where we are located is also a viable business opportunity both technically and economically and very interesting in the market. You can find this information in the following table .

Economics results of RO membrane recycling study in EUR.

TRADING ACCOUNT

YEAR 0

YEAR 1

YEAR 2

 YEAR 3

YEAR 4

 YEAR 5

Bills

500

300

300

350

350

Variation of existences

0

0

0

0

0

Shopping

5000

3000

3000

3500

3500

External services

1000

500

500

600

600

Personal expenses

30000

30900

31827

32782

33765

Amortization

4000

4000

4000

4000

4000

Financial expenses

100

100

100

120

120

Other expenses

350

250

200

200

200

Total

40950

39050

39927

41552

42535

Income

100000

100000

100500

121200

121800

Variation of existences

0

0

50

100

150

Sales (obsolete membranes)

10000

10000

10050

10100

10150

Other operating income

0

0

0

0

0

Other income

0

0

0

0

0

Financial income

0

0

0

0

0

Total

-80000

59050

60950

60573

79648

79265

Internal Rate of Returns (IRR)

73%

Also, the authors should do a bit more search in terms of the number of existing plants and their capacities. The data is significantly higher than mentioned in the article.

We have introduced more information about the number of existing plants and their capacities in the Atlantic islands to improve the data following the comments of the reviewer.

For more information, you can find the below documentation used for this study about the SWRO plants data base of the Canary Islands. This is shown in table 2 and Figures 4-5.

Figure 4.   Most significant seawater desalination plants (2019).

Figure 5. Most significant seawater desalination plants in the Canary Islands (Spain) (2019).

Table 2. SWRO Plants in Canary Islands.

Name of the Plant

Production (m3/d)

Consume (kWh/m3)

Island

Habitants per Plant

Cercado de Don Andrés

200

3.5

Lanzarote

Irrigation

Lanzarote III 1

10000

3.5

Lanzarote

10541

Lanzarote III 2

5000

3.5

Lanzarote

5271

Lanzarote III 3

5000

3.5

Lanzarote

5271

Lanzarote IV

20000

3.5

Lanzarote

21083

Lanzarote V

18000

2.4

Lanzarote

18975

Aeropuerto

700

3.04

Lanzarote

18327

Agua Park

30

3.04

Lanzarote

500

Apartamentos Ficus

60

3.5

Lanzarote

120

Apartamentos Puerto Tahiche

150

3.5

Lanzarote

300

Apartamentos Trebol

80

3.5

Lanzarote

160

Ercros

2500

3.5

Lanzarote

11057

Ercros

2200

3.5

Lanzarote

9731

Famara

350

3.5

Lanzarote

700

Hotel Golf y Mar

90

3.5

Lanzarote

180

Hotel Gran Meliá Salinas

400

2.61

Lanzarote

800

Hotel Playa Verde

250

3.5

Lanzarote

500

Hotel Teguise Playa

250

3.5

Lanzarote

500

La Galea

150

3.04

Lanzarote

300

Lanzarote Beach Club II

70

3.04

Lanzarote

140

Las Arenas. Costa Teguise

80

3.04

Lanzarote

160

Playa Roca

250

3.04

Lanzarote

500

Apartamentos Don Paco Castilla

320

2.61

Lanzarote

640

Apartamentos Sol Lanzarote

350

2.61

Lanzarote

700

Cdad Apartamentos CAMP

2.61

Lanzarote

Tourism

Holiday Land S.A.

3000

3.5

Lanzarote

6000

Hotel Fariones Playa

500

3.5

Lanzarote

1000

Hotel Playa Azul

300

3.5

Lanzarote

600

Hoteles Canarios S.A.

3.5

Lanzarote

Tourism

Iberhotel

3.5

Lanzarote

Tourism

Zorilla

40

3.04

Lanzarote

80

Hotel Jameos Playa

336

2.61

Lanzarote

672

La Santa Sport I

250

3.5

Lanzarote

500

La Santa Sport II

250

3.5

Lanzarote

500

Ria La Santa

400

3.5

Lanzarote

800

Apartamentos Son Boy Family Suites

500

3.04

Lanzarote

1000

Bungalows Atlantic Gardens

3.5

Lanzarote

Tourism

Costa los Limones S.A.

350

3.5

Lanzarote

700

Hotel Corbeta

3.5

Lanzarote

Tourism

Hotel Costa Calero

324

3.04

Lanzarote

642

Marina Rubicón

300

3.04

Lanzarote

600

Hotel Paradise Island

300

3.04

Lanzarote

600

Hotel Princesa Yaiza

500

3.04

Lanzarote

1000

Hotel Rubicón Palace

450

3.04

Lanzarote

900

Inalsa Sur 1

600

3.5

Lanzarote

1859

Inalsa Sur 2

1200

3.5

Lanzarote

3718

Inalsa Sur 3

3000

3.5

Lanzarote

9294

Janubio

3.04

Lanzarote

Tourism

Lanzasur Club

200

3.04

Lanzarote

400

Playa Blanca S.A.

3.5

Lanzarote

Tourism

Club Lanzarote

4500

3.5

Lanzarote

9000

Apartamentos Moromar

250

3.5

Lanzarote

500

Gea Fonds Numero Uno Lanzarote S.A.

3.5

Lanzarote

Tourism

Grupo Rosa

1000

3.5

Lanzarote

2000

Hipotels

300

3.5

Lanzarote

600

Hotel Corona

300

3.5

Lanzarote

600

Hotel Costa Calero S.L.

300

3.04

Lanzarote

600

Hotel Sunbou

500

3.04

Lanzarote

1000

Isla Lobos

100

3.04

Lanzarote

200

Leas Hotel S.A.

3.5

Lanzarote

Tourism

Niels Prahm

3.5

Lanzarote

Tourism

Occidental Hotel Oasis

250

3.04

Lanzarote

500

Playa Flamingo

200

3.04

Lanzarote

400

Tjaereborg Timesharing. S.A.

500

3.04

Lanzarote

1000

Empresa Mixta de Aguas de Antigua. S.L.

4800

3.04

Fuerteventura

11948

Grupo Turístico Barceló. S.L.

240

3.5

Fuerteventura

480

Aguas Cristóbal Franquis. S.L.

1200

3.5

Fuerteventura

2400

Anjoca Canarias. S.A.

3000

3.5

Fuerteventura

6000

Ramiterra. S.L.

3000

3.04

Fuerteventura

6000

Inver Canary Dos. S.L.

300

3.04

Fuerteventura

600

Suministros de Agua de La Oliva. S.A.

9000

3.04

Fuerteventura

17920

Consorcio Abastecimiento de Aguas a Fuerteventura

4000

3.04

Fuerteventura

7964

Parque de Ocio y Cultura (BAKU) 1

300

3.04

Fuerteventura

600

Parque de Ocio y Cultura (BAKU) 2

90

3.04

Fuerteventura

180

RIU Palace Tres Islas

100

3.5

Fuerteventura

200

RIU Oliva Beach

400

3.5

Fuerteventura

800

Nombredo. S.L.

500

3.5

Fuerteventura

1000

Consorcio Abastecimiento de Aguas a Fuerteventura

4400

3.5

Fuerteventura

20539

Puertito de la Cruz

60

3.5

Fuerteventura

120

Vinamar. S.A.

3600

3.5

Fuerteventura

7200

Fuercan. S.L. Cañada del Rio I

2000

3.5

Fuerteventura

4000

Fuercan. S.L. Cañada del Rio II

1000

3.04

Fuerteventura

2000

Fuercan. S.L. Cañada del Rio III

2000

3.04

Fuerteventura

4000

Club Aldiana

200

3.5

Fuerteventura

400

Erwin Sick

30

3.5

Fuerteventura

60

Esquinzo Urbanización II

1200

3.5

Fuerteventura

2400

Esquinzo Urbanización III

1200

3.5

Fuerteventura

2400

Hotel Sol Élite Los Gorriones 1

400

3.5

Fuerteventura

800

Hotel Sol Élite Los Gorriones 2

400

3.5

Fuerteventura

800

Stella Canaris I

300

3.5

Fuerteventura

600

Stella Canaris II

300

3.5

Fuerteventura

600

Stella Canaris III

250

3.5

Fuerteventura

500

Hotel H 10 Playa Esmeralda.

250

3.5

Fuerteventura

500

Hotel “Club Paraíso Playa"

300

3.5

Fuerteventura

600

Urbanización Costa Calma.

110

3.5

Fuerteventura

220

Urbanización Tierra Dorada.

120

3.5

Fuerteventura

240

Zoo-Parque La Lajita.

1300

3.5

Fuerteventura

500

Apartamentos Esmeralda Maris

120

3.5

Fuerteventura

240

Hotel H10 Tindaya

280

3.5

Fuerteventura

560

Aparthotels Morasol

80

3.5

Fuerteventura

160

Consorcio Abastecimiento de Aguas a Fuerteventura

36500

3.5

Fuerteventura

39382

Aeropuerto

500

3.5

Fuerteventura

15439

GranTarajal

4000

3.5

Fuerteventura

14791

Sotavento. S.A.

2925

3.5

Fuerteventura

5850

Arucas-Moya I

10000

3.5

Gran Canaria

45419

Granja experimental

500

3.5

Gran Canaria

Irrigation

Granja experimental

500

3.5

Gran Canaria

Irrigation

Comunidad Fuentes de Quintanilla

800

3.04

Gran Canaria

Irrigation

Granja experimental

500

3.5

Gran Canaria

Irrigation

Gáldar-Agaete I

3000

3.5

Gran Canaria

16199

Gáldar II

7000

3.04

Gran Canaria

37799

Agragua

15000

3.5

Gran Canaria

Irrigation

Guía I

5000

3.5

Gran Canaria

6962

Guía II

5000

2.61

Gran Canaria

6962

Félix Santiago Melián

5000

2.61

Gran Canaria

Irrigation

Las Palmas III

65000

3.5

Gran Canaria

307545

Las Palmas IV

15000

2.61

Gran Canaria

70972

BAXTER S.A.

100

3.5

Gran Canaria

200

El Corte Inglés. S.A.

300

3.5

Gran Canaria

3000

Anfi del Mar I

250

3.5

Gran Canaria

500

Anfi del Mar II

250

3.5

Gran Canaria

500

AQUALING

2000

3.04

Gran Canaria

4000

Puerto Rico

4000

3.04

Gran Canaria

8000

Puerto Rico I

4000

3.04

Gran Canaria

8000

Hotel Taurito

400

3.04

Gran Canaria

800

Hotel Costa Meloneras

300

3.04

Gran Canaria

600

Hotel Villa del Conde

500

3.04

Gran Canaria

1000

Bahia Feliz

600

3.5

Gran Canaria

1200

Bonny

8000

3.5

Gran Canaria

Irrigation

Maspalomas I Mar

14500

3.5

Gran Canaria

19572

Maspalomas II

25200

3.04

Gran Canaria

34016

UNELCO II

600

3.5

Gran Canaria

Industrial

Ayto. San Nicolas

5000

3.04

Gran Canaria

7608

Asociación de agricultores de la Aldea

5400

3.04

Gran Canaria

Irrigation

Sureste III

8000

3.5

Gran Canaria

133846

Aeropuerto I

1000

3.5

Gran Canaria

24791

Salinetas

16000

3.5

Gran Canaria

102424

Aeropuerto II

500

3.5

Gran Canaria

12396

Hoya León

1500

3.5

Gran Canaria

Irrigation

Bco. García Ruiz

1000

3.5

Gran Canaria

Irrigation

Mando Aéreo de Canarias

1000

3.5

Gran Canaria

3000

UNELCO I

1000

3.5

Gran Canaria

Industrial

Anfi del Mar

1500

3.04

Gran Canaria

3000

Norcrost. S.A.

170

3.04

Gran Canaria

340

Adeje Arona

30000

3.04

Tenerife

126728

Gran Hotel Anthelia Park

3.04

Tenerife

Tourism

La Caleta (Ayto. Adeje)

10000

3.04

Tenerife

20000

UTE Tenerife Oeste

14000

2.16

Tenerife

40000

Hotel Sheraton La Caleta

3.04

Tenerife

Tourism

Hotel Gran Tacande

3.04

Tenerife

Tourism

Hotel Rocas de Nivaria. Playa Paraíso

3.04

Tenerife

Tourism

Hotel Bahía del Duque. Costa Adeje

3.04

Tenerife

Tourism

Siam Park

3.04

Tenerife

Tourism

Tenerife-Sol S. A.

3.04

Tenerife

Tourism

Hotel Conquistador. P. de Las Américas

3.04

Tenerife

Tourism

Arona Gran Hotel. Los Cristianos

3.04

Tenerife

Tourism

Bonny S.A.. Finca El Fraile.

3.04

Tenerife

Tourism

El Toscal. La Estrella (C. Regantes Las Galletas)

3.04

Tenerife

Tourism

Complejo Mare Nostrum. P. Las Américas

3.04

Tenerife

Tourism

Hotel Villa Cortés

3.04

Tenerife

Tourism

Buenavista Golf S.A.

3.04

Tenerife

Tourism

Rural Teno

3.04

Tenerife

Agrícola

Ropa Rent. S.A. (P.I. Güímar)

3.04

Tenerife

Industrial

Unelco

600

3.5

Tenerife

Industrial

I.T.E.R. Cabildo de Tenerife

14

3.5

Tenerife

Industrial

C.T. en P.I. de Granadilla

3.5

Tenerife

Industrial

Bonny S.A.. Finca El Confital.

3.5

Tenerife

Irrigation

Polígono Industrial de Granadilla (portátil)

3.5

Tenerife

Industrial

UTE Desalinizadora de Granadilla

14000

3.04

Tenerife

50146

Guia de ISORA Hoya de la leña

3.5

Tenerife

Tourism

Club Campo Guía de Isora. Abama

3.5

Tenerife

Tourism

Hotel Meliá Palacio de Isora. Alcalá.

3.5

Tenerife

Tourism

Loro Parque

3.5

Tenerife

Tourism

Santa Cruz I

20000

3.04

Tenerife

204856

Recinto Portuario Santa Cruz (portátil)

3.04

Tenerife

Industrial

CEPSA

1000

3.04

Tenerife

Industrial

Hotel Playa la Arena

3.04

Tenerife

Tourism

Hotel Jardín Tecina

2000

3.04

La Gomera

4000

La Restinga

500

3.5

El Hierro

297

La Restinga

1200

3.04

El Hierro

712

El Cangrejo

1200

3.04

El Hierro

2478

El Cangrejo

1200

3.04

El Hierro

2478

El Golfo

1350

3.04

El Hierro

4093

The figures should be placed closest to the place where they are mentioned in the article. Also, I also feel that figures are not indicative of the context.

Thanks, you are right, we have modified the figures in order to be placed closet to the right position in the article agreed with the context. Therefore, we have done it too and the paper has been improved.

The data on circular economy is probably extracted from literature and hence authors should specify the source.

You are right. We have introduced all the references extracted from literature and also source of every table and figure. Thanks for this comment due to increase much more the quality and references of the paper.

The authors should add some latest references as well.

Ok, we have increased the number of references adding the lasted ones of the last years, improving much more the references and discussion of the manuscript. For instance the following:

[23] Will Lawler a, Zenah Bradford-Hartke a, Marlene J. Cran b, Mikel Duke b,c, Greg Leslie a, Bradley P. Ladewig d,e, Pierre Le-Clech a,⁎ Towards new opportunities for reuse, recycling and disposal of used RO membranes Desalination 299 (2012) 103–112

[24] Rodriguez J.J., Jimenez V., Trujillo O. and Veza J.M., Reuse of RO membranes as a filtration stage in advanced wastewater treatment, Desalination, 150 (2002) 219-226.

[25] Govardhan B., Fatima S., Madhumala M. and Sridhar S. Modification of used commercial RO membranes to nanofiltration modules for the production of mineral‑rich packaged drinking water. Applied Water Science (2020) 10:230 https://doi.org/10.1007/s13201-020-01312-1

[26] Zhou, J. Chang, V. W. C. and A. G. Fane (2011): Environmental life cycle assessment of brackish water RO desalination for different electricity production models”, Energy and Environmental Science, 4, 6, 2267-2268.

[27] J. V. Reboso, F. N. Toyos, J. R. S. Rámirez, and B. P. Suárez, “Application of RO to the regeneration of wastewater in the Southeast of the island of Gran Canaria (Spain),” Desalination and Water Treatment, vol. 208. 2020, doi: 10.5004/dwt.2020.26449

[28] Garcia R., Nanofiltration and ultrafiltration membranes from end-of-life RO membranes. A study of recycling. Thesis. Universidad de Alcalá de Henares (2017).

[29] Sohum K. Patel, Cody L. Ritt, Akshay Deshmukh, Zhangxin Wang, Mohan Qin, Razi Epsztein, Menachem Elimelech. The relative insignificance of advanced materials in enhancing the energy efficiency of desalination technologies. Energy Environ. Sci., 13 (6) (2020) 1694-1710.

[30] Jafari, M.; Vanoppen, M.; van Agtmaal, J.M.C.; Cornelissen, E.R.; Vrouwenvelder, J.S.; Verliefde, A.; van Loosdrecht, M.C.M.; Picioreanu , C. Cost of founling in full-scale reverse osmosis nanofiltration installations in the Netherlands. Desalination 2021, 500, 114865.

[31] Jafari, M.; Vanoppen, M.; van Agtmaal, J.M.C.; Cornelissen, E.R.; Vrouwenvelder, J.S.; Verliefde, A.; van Loosdrecht, M.C.M.; Picioreanu , C. Cost of founling in full-scale reverse osmosis nanofiltration installations in the Netherlands. Desalination 2021, 500, 114865.

[32] Tavares, T., Tavares, J, et al. Assessment of Processes to Increase the Useful Life and the Reuse of Reverse Osmosis Elements in Cape Verde and Macaronesia. Membranes. 2022, 12, 613. https://doi.org/10.3390/membranes12060613

Overall, I think, the authors should rewrite the entire manuscript in a systematic and organized manner.

Following the kind advise of the reviewer we rewrote the manuscript in a systematic and organized manner more under stable and flexible for reading, improving much more the structure, contents and references. We have done a big effort in this way to publish this paper accepting all the comments and instructions which have been considering and done. Thanks so much.

That said, I think, the research topic is very relevant and it is important to carry out systematic research.

Thanks so much again for your opinion in this way and strong support to improve a lot our manuscript and to publish it.

Reviewer 3 Report (New Reviewer)

Comments to the Author:

In this paper, the reverse osmosis (RO) membrane is treated by pyrolysis and the RO elements are recovered for fuel production, which is in line with the circular economy effect. However, there are some important points which should be addressed before a further consideration. I suggest major revision of the manuscript based on the following comments:

1. The abstract content is too similar to the first paragraph of the article. Please revise and simplify the abstract content.

2. In line 38, "As RO (RO) desalination to obtain such water", the complete expression of the RO outside the parentheses should be indicated.

3. The gases mentioned in the article, such as "CO2, H2", shall be regulated, such as CO2.

4. Please redraw Figure 3 to make it as beautiful and simple as possible.

5. The conclusions lead to a simple summary with positive notes, but the final paragraph is lacks logic. If possible, please revise the conclusion to make it simple and clear.

Author Response

Author's Notes to Reviewer.

In this paper, the reverse osmosis (RO) membrane is treated by pyrolysis and the RO elements are recovered for fuel production, which is in line with the circular economy effect. However, there are some important points which should be addressed before a further consideration. I suggest major revision of the manuscript based on the following comments:

Thanks so much for your comments and kind support in this review, due to it helped us to improve a lot our manuscript. We strongly followed your instructions and good advice.

  1. The abstract content is too similar to the first paragraph of the article. Please revise and simplify the abstract content.

You are right. We revised and simplified the abstract content and finally we eliminated any duplication between the abstract and this paragraph of the article.

  1. In line 38, "As RO (RO) desalination to obtain such water", the complete expression of the RO outside the parentheses should be indicated.

We agree with you. We completed and indicated accordingly the expression of RO outside the parenthesis.

  1. The gases mentioned in the article, such as "CO2, H2", shall be regulated, such as CO2.

OK, we changed it following the good advice of the reviewer.

  1. Please redraw Figure 3 to make it as beautiful and simple as possible.

Ok, we redrew figures 3 and also figure 4 for being more beautiful, simple and under stable.

  1. The conclusions lead to a simple summary with positive notes, but the final paragraph is lacks logic. If possible, please revise the conclusion to make it simple and clear.

You are right. We revised the conclusions to make them simpler and clearer than before, following the comments of the reviewer.

Thanks so much again for your kind comments and strong support to give us the opportunity to improve a lot our manuscript and to publish it.

Round 2

Reviewer 1 Report (New Reviewer)

The results and discussion section should be further improved with recent citations 

Author Response

Dear reviewer,

Thanks so much for your support in this manuscript and to help us to improve it. Regarding your comments below we can tell you the following:

The results and discussion section should be further improved with recent citations

We agree with you, and we have introduced recent citations to improve results and discussion section. Finally, we introduced 10 new recent citations more updated in the results and discussion section.

Thanks so much again for your opinion about this study, strong support to improve much more our manuscript and to publish it.

Best regards.

The authors.

Reviewer 2 Report (New Reviewer)

The topic is of significant importance. However., the authors are requested to conduct an economic analysis to justify conclusions.

Please make sure that if a figure is borrowed from either web or publication, please provide attribution.

Also, please highlight the novelty of this investigation.

Please have someone read the manuscript for grammar.

Author Response

Dear reviewer,

Thanks so much for your support in this manuscript and to help us to improve it. Regarding your comments below we can tell you the following:

The topic is of significant importance. However., the authors are requested to conduct an economic analysis to justify conclusions.

Thanks so much for your opinion about the topic and to help us to improve it. We agree with you and following your comments we have introduced an economic analysis to justify the conclusions at the end of the results section with 10 new recent references updated, as you can see in the new manuscript modified. Moreover, we have also introduced an appendix for more economic information and real data about costs.

Please make sure that if a figure is borrowed from either web or publication, please provide attribution.

Yes, you are right, mostly figures are made by us, and we have introduced all the references related to them. Therefore, you can find all the references used from us for helping in this study. In fact, we have introduced 10 new recent references used in our study and we have updated them.

Also, please highlight the novelty of this investigation.

Ok, we have highlighted the novelty of the investigation introducing new paragraphs in the abstract, introduction and conclusions sections of this manuscript to show the significant importance of this study.

Please have someone read the manuscript for grammar.

Of course, we will share the manuscript with a professional to improve the grammar. Thanks so much again for your opinion about this study, strong support to improve much more our manuscript and to publish it.

Best regards.

The authors.

Reviewer 3 Report (New Reviewer)

(1) In Figure 1, there is only the polymer structure, and other information including chemical tolerance, membrane manufacturing method, operating conditions and performance is absent.

(2) There is only figure caption for Figure 2, and detailed figure is absent.

(3). In L 188-192, how to find image 1, 2? In addition, this paragraph is very hard to understand. Thus, it needs to be revised.

(4) The authors need to improve the logicality of whole manuscript.

Author Response

Dear reviewer,

Thanks so much for your support in this manuscript and to help us to improve it. Regarding your comments below we can tell you the following:

(1) In Figure 1, there is only the polymer structure, and other information including chemical tolerance, membrane manufacturing method, operating conditions and performance is absent.

Thanks so much for your review of this manuscript and kind support to improve it. We agree with you, we have introduced more information after figure 1, not only about polymer structure but also chemical tolerance, membrane manufacturing method, operating conditions, and performance. We have included a new figure 2 including much more information about the membrane composition, chemical tolerance, operating and performance conditions.

(2) There is only figure caption for Figure 2, and detailed figure is absent.

You are right. We have introduced the new figure 2 and several paragraphs, introducing more contents and text before and after figure 2, with all the details about the RO membrane to improve much more the manuscript regarding this issue.

(3). In L 188-192, how to find image 1, 2? In addition, this paragraph is very hard to understand. Thus, it needs to be revised.

Sorry, you are right, we have revised and modify this content to be more understandable and correct grammatically. We have introduced several changes to be easier to understand than before and we have explained more the figures 1 and 2 introducing several paragraphs with details increasing much more the information shown in the manuscript.

(4) The authors need to improve the logicality of whole manuscript.

Thanks so much again for your opinion in this way and strong support to improve a lot our manuscript. Following your kind comments and good advice we have also improved the logicality of whole manuscript, organized the information, and introduced new one. Due to this, finally we have finished the manuscript much more complete, understandable, and easier to follow up than before.

Best regards.

The authors.

Round 3

Reviewer 2 Report (New Reviewer)

The authors have made several changes and it is better than before. It can be accepted after English check by the publishers.

This manuscript is a resubmission of an earlier submission. The following is a list of the peer review reports and author responses from that submission.

Round 1

Reviewer 1 Report

The authors address the recycling of membrane materials for the circular economy. The manuscript is poorly written and poorly structured. Moreover, the manuscript does not provide adequate information to support the conclusions. The concerns are as follows:

1. In the abstract, the authors describe the importance of water, but then abruptly change into reverse osmosis membrane recycling. From the reviewer's perspective, the structure of the abstract is weirdly constructed. A thorough reconstruction is necessary for readability. 

2. At the end of the abstract, the authors mention that "This is a technically and economically viable business opportunity with a promising future in today's recycling market as discussed in the article." There is no evidence in the body of the manuscript to support this claim.

3. The Introduction section is also not well written with a nice storyline, major improvement is necessary.

4. In section 2, the authors mentioned that this study is carried out to provide an initial assessment of their feasibility. The authors should provide metrics for such assessment, not only just by pointing out available options without further analysis and reliable metrics, i.e., financial, social, or other KPIs.

4. The claims in the conclusions are made without solid evidence in the body of the manuscript. The authors should put more effort into producing the evidence before these claims.

5. There are several typographical errors and paragraph structure issues that should be addressed by the authors.

Author Response

Reviewer.

The authors address the recycling of membrane materials for the circular economy. The manuscript is poorly written and poorly structured. Moreover, the manuscript does not provide adequate information to support the conclusions. The concerns are as follows:

Dear reviewer,

Firstly, thanks so much for your high support in this manuscript. Therefore, we strongly followed your comments and we got to improve the paper much more.

  1. In the abstract, the authors describe the importance of water, but then abruptly change into reverse osmosis membrane recycling. From the reviewer's perspective, the structure of the abstract is weirdly constructed. A thorough reconstruction is necessary for readability. 

You are right, we reconstructed the abstract following the reviewer advice to improve the structure of this one for readability.

  1. At the end of the abstract, the authors mention that "This is a technically and economically viable business opportunity with a promising future in today's recycling market as discussed in the article." There is no evidence in the body of the manuscript to support this claim.

Ok, we eliminated this sentence in the abstract to avoid any misunderstanding and improved the structure of the abstract.

  1. The Introduction section is also not well written with a nice storyline, major improvement is necessary.

You are right, we also improved the written of the Introduction to be it more understandable and comprehensive.

  1. In section 2, the authors mentioned that this study is carried out to provide an initial assessment of their feasibility. The authors should provide metrics for such assessment, not only just by pointing out available options without further analysis and reliable metrics, i.e., financial, social, or other KPIs.

Ok, we have introduced more information regarding this issue in section 2 of the manuscript to improve much more it. Anyway, we eliminated this sentence and improved the content of this part to avoid any misudertanding.

  1. The claims in the conclusions are made without solid evidence in the body of the manuscript. The authors should put more effort into producing the evidence before these claims.

Ok, we have made an effort to introduce more solid evidence in the body of the manuscript to support the conclusions commented.

  1. There are several typographical errors and paragraph structure issues that should be addressed by the authors.

Yes, we corrected the typo errors and paragraph structure to improve the manuscript in general.

Thanks so much again for your help to improve our manuscript very much.

Reviewer 2 Report

 This study evaluate the processes to increase the useful life and potential recycling of reverse osmosis membranes and components. The introduction of the research background is too general and does not focus on the research front of this study. The overall study lacks theoretical and experimental data support. It is difficult to judge the accuracy of the conclusion.

Author Response

Reviewer.

This study evaluate the processes to increase the useful life and potential recycling of reverse osmosis membranes and components. The introduction of the research background is too general and does not focus on the research front of this study. The overall study lacks theoretical and experimental data support. It is difficult to judge the accuracy of the conclusion.

Dear reviewer,

Firstly, thanks so much for your high support in this manuscript. Therefore, we strongly followed your comments and we got to improve the paper much more.

You are right, we have improved the Introduction to be it more understandable and comprehensive.

We also have introduced reliable metrics, including financial, social and others in the manuscript, to improve the theorical and experimental data.

Moreover, we have tried to introduce more solid evidence in the body of the manuscript to support the conclusions commented

Thanks so much again for your help to improve our manuscript

Round 2

Reviewer 2 Report

1. How to evaluate the economics of the method in this study?

2. The experimental results and analysis evaluation of this study are too few to judge the accuracy of the conclusion.